# Are Innovation and Creative Districts New Scenarios for Sustainable Urban Planning? Bogota, Medellin, and Barranquilla as Case Studies

Ana Elena Builes-Vélez [1,*], Lina María Escobar [1] and Claudia Villamil-Mejia [2]

1   School of Architecture and Design, Universidad Pontificia Bolivariana, Medellin 050031, Colombia; linam.escobar@upb.edu.co
2   Independent Researcher, 46022 Valencia, Spain; a.villamilmejia@gmail.com
*   Correspondence: ana.builes@upb.edu.co

**Abstract:** Creative and innovation districts are focused on boosting local economies. However, they also pay attention to the global scale since this local identity of the Orange Economies gives them added value and competitiveness globally, as well as international projection and visibility of products, services, and new technologies associated with creativity and innovation. Thus, this study reviews three case studies of the cities of Barranquilla, Bogota, and Medellin in Colombia, seeking to characterize the creative and innovation districts. The methodology used is strictly qualitative, resorting to the characterization of the polygons and the analysis of public policies. This study describes the general aspects of creative and innovation districts in Colombia. Then, it identifies the forces for CD and ID development in three Colombian cities and the drivers for urban sustainable transformation. The main purpose of this study is to understand if innovation and creative districts are new scenarios for sustainable urban planning.

**Keywords:** orange economy; innovation districts; creative districts; public policy

## 1. Introduction

In the last three decades of the 20th century, cities and metropolitan regions, mainly in post-industrialized economies, have undergone an important process of economic transformation due to the consolidation of services as the primary source of economic development, especially knowledge-intensive services, and services in creative and cultural sectors or what is commonly known in our context as the "Orange Economy". Orange Economy (OE), according to the National Ministry of Culture, is a tool for social development that integrates the arts, cultural heritage, cultural industries, and functional creations toward the social enterprise. Characteristics of an OE focus on human creativity and imagination, centered on art, culture, and entrepreneurship [1]. As these economic sectors involve the transformation of individual talent, unlike other traditional production economies, they require constant social interaction and, therefore, constant spatial interaction. The advantages offered by localization have thus become a highly decisive factor for the growth of creative and knowledge economies since economic globalization and telecommunications growth have encouraged the mobility of talent and capital, making local geographical advantages more decisive [2]. Regional competitiveness is a consequence of this process. To the traditional factors that companies look for in order to locate, such as the size of the labor market and communication infrastructures, are now added specific territorial conditions, such as institutional organization and the capital existing in people, either due to their behavior in society-social capital or their skills and knowledge-cognitive capital.

All these aspects require new approaches based on the forms of institutional organization and governance. They also need approaches based on the policies implemented in territorial planning, especially in metropolitan areas where sectors are transformed due

to various reasons. For example, due to multiple political, economic, or social reasons, they turn into urban projects and zones that change their soil aptitude and acquire new characteristics. Therefore, the drivers of change studied include public policies, road structure transformation, soil overexploitation, climate change, pollution, and architectural obsolescence. These are common elements in the three cities' creative and innovation districts under analysis.

## 2. Literature Review

### 2.1. Creative and Innovation Districts

The creative districts (CDs) are part of these city sectors and are scenarios for cultural expression. According to Marques and Richards [3], they have been increasingly integrated into the cities and become the heart of artistic life, often linked to bohemian life. Such artistic or cultural expressions of tradition or forefront accompany the districts under transformation, and they (the districts) take the activities that take place there and turn them into their identity. Likewise, CDs and innovation districts (IDs) are part of the municipal administrations' plans; when large areas fall into disuse, new actors emerge to intervene.

The CDs are:

Geographically delimited spaces where culture, creativity, and entrepreneurship converge and function as areas of economic, social, and cultural development, consolidating scenarios for urban renewal, job creation, and the creation, production, distribution, exhibition, commercialization, and consumption of cultural and creative goods and services [3].

In addition, when various activities appear in the city's transformations, it is important to create urban plans that allow the development of adequate conditions for the new uses that arise and, at the same time, contribute to territorial planning. Consequently, IDs are part of recent knowledge, entrepreneurship, and technology trends.

Particularly, territorial planning in Colombia, governed by the Territorial Development Law 388 of 1997, grants municipal planners various instruments for managing urban growth, where urban development strategies are clearly integrated. However, as previously stated, multiple capitals located in the territory are necessary to grow and expand these economic activities within cities. Hence, there is a need to approach public policies oriented to the development of districts from a critical point of view to house these economic activities in the context of Colombian cities.

Furthermore, creative and knowledge-based economic activities frequently establish economic relations with other industrial economic sectors and/or non-specialized services, mainly because they are suppliers of goods and services to the latter. Therefore, a public policy to promote these knowledge and creative sectors must focus not only on the promotion of clusters in these economic activities but also on the promotion of economic growth as a whole, backed by adequate urban development and based on the generation of communication and other support infrastructures [4]. Ferreiro-Seoane et al. argue that "Creative and cultural activities, more recently encompassed in the Orange economy, (...) have been attributed great potential as engines of economic growth and sustainable development" [5] since the services and products that result from creative and cultural sectors are constantly innovating and thinking in a more sustainable city, promoting healthier relations with their immediate context.

According to Chica-Mejía et al. [6], creative cities constitute a special synthesis of the new agglomeration economies resulting from the change in production models that require the concentration of qualified human resources. The re-configuration of the territories into clusters or districts has shown different approaches to urban morphology from urban form and function [7]; due to their spatial structure, cities with clustering structures allow the creation of creative and innovative networks [8,9]. In Colombia, the economic relevance of creative industries in employment is small and grew little between 2008 and 2017 [10]. Specialties produced and driven by these approaches share similar and complementary features. Firstly, place matters as a principal and a criterion, and physical and physical–spatial

values such as form, meaning, and activity are derived. Secondly, creative and innovative industries, actors, and processes are sensitive to the characteristics of the place since some factors and urban configurations favor the emergence, consolidation, and success of districts [11]. According to Chang et al., these configurations allow place attachment and restitution of social environments that strengthen creative and innovative networks [12].

## 2.2. Sustainability and Urban Planning

Human activities are linked to urban areas. The majority of the world's population lives in cities that, due to the rapid growth of the population, are forced to rethink their urban planning policies and strategies. Water management, clean air, sanitation, and excellent transportation links are some of the issues urban planners face nowadays. According to Puchol-Salort et al. [13], "While managing the effects of urban growth represents a significant challenge, it also provides an opportunity to rethink how we plan and design our urban spaces to support sustainability, both in terms of new urban development projects and in retro-fitting existing urban spaces". Urban development projects must include a plan to articulate grey infrastructure with green, blue, and public space, rethinking interactions between build space, vegetation, and any types of natural surfaces to contribute to the sustainable development of the cities.

Integrating sustainability into urban planning is complex because many factors act and interact in different systems within the cities, including "social, built and natural systems which intersect and interact in numerous complex ways that represent new urban pressures (land use change, air and soil contamination, green space reduction, etc.) and critical challenges (social equity, well-being, administrative cooperation, etc." [13]. Achieving sustainable urban development can be difficult, but not impossible, and requires system thinking [14], understanding the city as a system of systems interconnected and naturally changing due to those interactions. Also, understanding the interconnections between city districts and spaces that reflect the city's identity, in this case, creative and innovation districts that promote alternative models for development, including sustainability and Orange Economy in the equation.

## 2.3. Orange and Creative Economy and Sustainability

The creative economy has become an alternative development model. Colombia, like many other countries in Latin America and the Caribbean, is seeking to build policies that promote creative and cultural-based companies to favor economic growth and sustainable development. Cities like Bogota and Medellin have bet on clustering cultural and creative activities and declaring creative and innovative districts in order to connect and regroup stakeholders of these sectors to enhance sustainable relationships and develop new ways of inhabiting urban space.

The UN has pointed out the role of culture and creativity in sustainable development in conferences and agreements, but it was not until 2015 that culture was introduced into the 17 SDGs within the 2030 Agenda. The new global agenda places culture as a relevant sector for development, in which cultural and creative industries play an important role in producing new technologies or creative ideas and in generating non-monetized social benefits [15]. In this context, creative and culture industries have proven to have a positive impact on the development of sustainable cities and inclusive societies, since these industries help build a shared sense of cultural identity and cultural values help strengthen social cohesion [16]. The creative, cultural, and innovative industries promote interconnectedness within cities and build strong social relationships that strengthen economic and social development possibilities through Orange and creative economy activities.

## 3. Materials and Methods

This article is based on a research project conducted in Medellín, Bogotá, and Barranquilla between 2020 and 2022. The methodological approach for this research was qualitative and quantitative, focusing on qualitative information to characterize the poly-

gons in each case study. In the three cities, researchers attended community meetings and public demonstrations, interviewed community leaders and neighborhood associations, and attended various fieldwork observations, in which researchers collected information using multiple research tools that were selected due to the type of information needed to obtain from the field. Also, national, regional, and local public policies related to the districts were traced in order to identify which ones were spontaneous and which were the result of public policies.

A general characterization of the polygons was made, including their location, extension in hectares, main uses, approximate population, type of urban treatment, and original situation, see Table 1. Subsequently, analysis maps were prepared with the following categories: heights, land uses, and the density of commercial, residential, and industrial uses of the three cities.

**Table 1.** Primary data collection plan.

| Stakeholder | Data Collection Instruments |
| --- | --- |
| Mixed | Characterization of polygons. Participant observation and review of the secondary literature on the districts and their respective localities. Cartographic analysis. |
| Institutional | Micro-ethnographies and participant observation. Cartographic analysis. |
| Community-based | Micro-ethnographies and participant observation. Cartographic analysis. |
| Private | Micro-ethnographies and participant observation. Cartographic analysis. |

Elaborated by researchers.

A matrix of drivers of transformation was developed in order to approach the understanding of the innovation and creative districts' urban phenomenon. This enabled the establishment of parameters for a later comparison of the urban phenomena of the spontaneous and planned districts. Six basic drivers of change were identified: public policy, transformation of road infrastructure, overexploitation of the soil, climate change, pollution, and architectural obsolescence. The direct causes, impacts, social, economic, and physical implications, together with the actors and their roles, were identified for each driver of transformation.

## 4. Results

### 4.1. General Aspects of Colombian's Creative and Innovation Districts

In Colombia, the creative and cultural economy has received more attention from the government in the last decade as an agent of social, economic, and sustainable development. In this sense, it has been advocated to implement public policies that allow for sustainability and growth in the field of cultural public policies. This economic sector groups together those productive sectors oriented towards "the creation, production and commercialization of goods and services based on intangible contest of cultural nature" [17]. Thus, the creative economy integrates productive activities associated with creativity and intellectual capital as "generators of added value and development" [17].

Public policies for the cultural sector in Colombia have increasingly focused on strengthening the sustainability of the creative economy as a public problem and a driver of socioeconomic development [18]. This strengthening has included territorial strategies such as developing creative or cultural districts aimed at strengthening the governance of the low articulation and synergy of public, community, private, and institutional actors of the creative economy ecosystems at the local level.

A fundamental cultural policy strategy within the framework of the Integral Plan for Orange Economy (Plan Integral de Economía Naranja—PIEN) has been the implementation

of the Orange Development Areas (AND), a term adopted in national legislation for creative or cultural districts. The AND have aimed to strengthen the sustainability of the cultural economy through "the creation of geographic spaces that operate as centers of economic and creative activities, delimited and recognized through territorial planning instruments" [18], managed through the joint articulation of public, private, and civil society actors. The cultural districts have thus raised the centrality of advocating for a "networked mode of governance that transcends hierarchies and sectors, is highly collaborative and responds to the concerns of citizens and society" [18].

At the national level, according to the PIEN, the Orange Economy can be grouped into three main categories, see Table 2: (i) the arts and heritage sector, which includes the visual, performing and performing arts, cultural tourism, cultural heritage, and education in arts and culture; (ii) cultural industries: publishing, phonographic, audiovisual, among others; and (iii) functional creations, new media, and content software.

**Table 2.** Classification of creative and cultural industries based on Orange Economy in Colombia.

| Type | Sub-Type |
| --- | --- |
| Arts and Heritage | Visual Arts<br>Performing Arts and Shows<br>Cultural Tourism and Material and Unmaterial<br>Cultural Heritage<br>Education in Arts, Culture, and Creative Economy |
| Cultural Industries | Editorial<br>Phonographic<br>Audiovisual<br>News Agencies and Other Services of Information |
| Functional creations, New Media, and Software | Digital Media and Software<br>Design<br>Advertising |

Elaborated by the DNP with information from Ministerio de Cultura. 2021.

In Colombia, a relevant cultural policy strategy to strengthen the creative economy at a territorial level has been the creation of CDs and IDs. These districts are a type of cluster, understood as "concentrations of interconnected companies and institutions of a particular sector" [19], which promote competitiveness. Within the framework of the PIEN, ADNs are spaces that operate as centers of economic and creative activity that strengthen the creative economy at the territorial level, promote urban renewal, foster employment, and articulate the actors of the cultural ecosystem. It is important to note that there are two typologies of creative districts or ADNs established within the framework of the PIEN and PPDECC public policy, according to their differential mode of creation and management. On the one hand, induced districts are those cultural spaces configured from policies that have "the purpose of transforming unused spaces or depressed areas into new engines of economic, social and cultural development" [20]. These districts respond to national or local government-driven intervention to transform an urban space. On the other hand, spontaneous districts are defined as "agglomerations of economic activities associated with the cultural economy that arise naturally" [20] and usually have a more community origin associated with local, civil, and organizational actors.

Based on the understanding of the CD and ID as a cultural policy of urban scope, which seeks the re-configuration of the urban territory from strategies of delimitation and sustainable territorial planning, as well as from the strengthening of the governance of the cultural sector, urban governance is understood as "the relationship that is woven between the actors involved in the formulation [and management] of public policies focused on the development of cities" [21]. Bogota, as the capital, is the city with the majority of CDs and IDs in the territory.



*4.2. Bogota Case Study. Characterization of the Creative and Innovation Districts and the Orange Development Areas from the General Framework of the Public Policy*

Bogota, as a capital district, aims to establish a guiding regulatory framework for the creation and consolidation of Orange Development Areas (ADN) by guiding processes of social impact within the intervened areas under private or local government investment and the reception of direct foreign investment such as those coming from ECLAC, IDB, among others. The Plan Decenal de Cultura Bogotá D.C. 2012–2021 [Ten-Year Culture Plan for Bogotá D.C.] [22] proposes strategies to lead processes for the integration of culture as one of the pillars of sustainable economic development of the city. It gathers proposals set forth in Agenda 21 for culture, the democratization of knowledge, and artistic, cultural, and cultural heritage practices. According to data from the National Administrative Department of Statistics (DANE for its Spanish acronym), the orange economy represents USD 1.03 billion in investment for Bogota, with higher growth in cultural sectors such as music (26.7%) and performing arts (22.6%), thus concentrating one of the most significant value chains of this sector in the country. At the district level, the ADNs are outlined for their economic development based on knowledge, the smart and digital city, the establishment of business–academia relations, and productivity supported by ICTs that allow clustering specialized environments in sustainable environments [3,23].

Eleven polygons are recognized for Bogota, of which Bronx D.C., Zona F, and San Felipe have been declared as priority areas that consider intervention plans to be consolidated as Innovation Districts [24]. This means "geographic areas where institutions and forefront anchor companies are grouped and connected with start-ups, business incubators, and accelerators. They are also physically compact, transit-accessible, and technically wired, and offer mixed-use housing, offices and stores" [25]. As a result of the layout of the polygons in the territory, it can be seen that the city concentrates its location on two important axes that correspond to arterial roads connecting the north–south and west–east.

From this scenario and based on Patterson, Schulz, Vervoot, and Van der Hel's [26] definition of the term territory transformation, which implies fundamental changes in the structure, functions, relationships, and cognitive aspects of socio–technical–ecological systems, which result in new patterns of interaction, whether they are incremental and carefully planned and performed by (often political) actors [27], or from emergent properties of large-scale political–economic forces and social mobilization [28]. These forces are recognized as drivers of transformation that are classified from the three approaches proposed by Scoones et al. [29], these being:

El Bronx D.C.: It has a structural approach, as induced areas with fundamental changes where political and economic processes associated with interests that perpetuate the conditions established by Decree 201 of 11 April 2019 that adopts the Partial Plan for Urban Renewal and Decree 145 of 2012, and of the National Development Plan 2018–2022 Pacto por Colombia, pacto por la equidad prevail [30].

This sector is characterized by the concentration of the largest number of heritage buildings in the city's historic center, which dominate the morphological structure with remnants of plots that make up a large urban void to be redefined. There is also a high architectural obsolescence resulting from illegal occupation by street dwellers, micro-trafficking, and informal trade. This has led to a scenario of marginality and degradation of urban land with a high structural deterioration at risk of collapsing.

La Candelaria (Spontaneous) and San Felipe (Mixed): enabling approach, as areas that emerge in the city and manage to consolidate either in the collective memory or through associativity of collectives located in the same territory. This approach focuses on promoting human agency, values, and capacities necessary to manage uncertainty collectively and identify and adopt paths toward desired futures. It describes in its essence the spontaneous or mixed areas, which present asymmetries in power and social injustice, causing artisans, population minorities, and formal or informal emerging economies to gather and organize despite the absence of political structure.

Although they produce changes in functionality and increase the demand for resources, they promote the improvement of local production through the exchange of knowledge in situ and the emergence of spaces for hackers and creators for innovation based on the roots, something common for sustainable local economies. Among the aspects to be controlled are the overexploitation of land use, the increase in noise, visual and atmospheric pollution, mobility complications due to increased demand and low capacity of road infrastructure and transportation systems, and the oversupply of commercial dynamics.

### 4.3. Case Study Medellin: Characterization of ADNs from the General Framework of Public Policy

Medellin has become one of the cities with the greatest creative potential in the country. Cultural events of local, regional, national, and international stature are held in the city, which is reflected in its economy. In a report prepared for the 2019 Business Meeting, the Chamber of Commerce reports that 6562 companies are registered in the city under the categories of activity of the orange economies, representing 3% of the city's GDP and creating about 33,000 formal jobs. However, it should be noted that this is an economic sector with a high level of informality, so much of the real data on those who are exercising creative and cultural economic activities in the city are outside the measurements and estimates produced by the Chamber of Commerce.

For its part, the Plan Intersectorial de la Economía Creativa de Medellín 2018–2030 [31] presents some preliminary data to characterize the sector. This highlights the added value of these industries, which shows a positive growth trend, and the formalization of companies, which has accelerated since 2007. It is also important to note that, of the 1695 companies registered, only nine are medium-sized and two are large, meaning that the sector is heavily concentrated in micro and small companies. In terms of local policies, the specific objectives of the Plan de Desarrollo Cultural de Medellín 2011–2020 [32] alluding to the creative and cultural industries, including guaranteeing the conditions for the circulation of cultural products, promoting creative dialogue through the use of information technologies, articulating education and culture at all levels and encouraging the creation of multi-sector cultural alliances [33].

Tracking the regulations allows for the extraction of information to perform the regulatory analysis related to the three Innovation and Creativity Districts in Medellin (El Poblado, Perpetuo Socorro, and Ruta N). Therefore, the identified resources include Decree 1483 of 2015 and Decree 1549 of 2016, which establish the MedellinInovation District, the Macroprojects, and the Special Management and Protection Plans (PEMP for its Spanish acronym) set by Ruta N as a strategic part of the innovation district [34]. Decree No. 2053 of 2015 and Resolution No. 201950108887 of 2019 designate and delimit the vocation of the Distrito Creativo del Perpetuo Socorro. Nevertheless, no regulations define, establish, or regulate Via Primavera as a design district [35].

In Medellin, the different districts have specific vocations, bringing us closer to the general definition of Ellie Cosgrave [36], who says that these innovation districts are growth centers in a city, which are stimulated by various factors. They are usually organically shaped and composed of start-up companies and creative industries clustered in large, connected, specialized, and economically diverse urban environments.

In this case, it is necessary to be aware of the consumption patterns emerging through the conformation of the city or specific spaces. As these patterns are accompanied by habits representing ways of life, they can modify the collective conscience and the social institution, regardless of their public or private character. It is observed that the districts, whether spontaneous or induced, are located in strategic points characterized by road infrastructure, predominantly with the urban renewal process, and its proximity to the city's downtown, positioning themselves as transformation projects favored by the economic diversity and the diversity of uses.

El Poblado: (Spontaneous) It is a commercial district in Medellin, where some social groups are symbolically organized with lifestyles supported or approved by socioeconomic practices such as "consuming leisure". Consequently, this increases social distances and as-

sociates the importance of being seen with obtaining prestige without establishing intimate links with other consumers in the space.

In the case of El Poblado, the buildings in the area that were once built to be residential have become fashionable commercial businesses physically configured that respond to the desire for exposure when consuming. Likewise, gastronomic venues such as cafes and restaurants have been established to enhance the idea of continuing to consume visibly: exhibiting the products obtained in those places or simply the ones being used. The phenomenon of exposure when consuming has improved the constant change of fashionable commercial enterprises and invigorated the way they emerge in the space.

Ruta N—Medellin Innovation: The innovation district (Induced) is a sector in the north of the city, which has undergone various interventions in recent years and is now transformed into an urban renewal cluster. Therefore, Medellin has chosen to promote the Innovation Center Ruta N model derived from the proposal of the Manzana de Emprendimiento (Entrepreneurship Block) contained in the Development Plan 2008–2011 to strengthen the knowledge-based economy [37] through the concentration of higher education, health and science, and culture facilities such as the Parque Explora and the Planetario. Additionally, it has important public spaces such as the Jardín Botánico and the Parque Norte.

These transformations have been promoted by architectural obsolescence, climate change, and pollution and include urban renewal interventions that involve an increase in vegetation cover and its impact on the ecological structure of the city. However, gentrification as a process of urban transformation is increasing real estate speculation, putting pressure on the resident population by increasing the cost of living in the sector.

Perpetuo Socorro: This is the creative district (Induced) located in the center of the city. It is a sector with a concentration of industrial uses where creative activities take place in the Bodega Comfama and the Mattelsa headquarters. However, the architectural detriment and pollution have led to the formulation of transformation processes through changes in land use that reduce the area of vegetation cover, an impact that leads to an increase in the heat island effect.

Moreover, it is important for this district to consider other aspects, such as heritage and restrictions per areas of influence of these assets in the sector. "If the crisis we face is an urban one, so is its solution. For all the challenges and tensions, they generate, cities are the strongest economic drivers the world has ever seen; the solution to the new urban crisis is more urbanism, not less." [38]. This considering that the objective established by the Mayor's Office of Medellin (18 November 2019) regarding the delimitation of this economic District is to enhance the capabilities of the community, renew it, strengthen the cultural institutions that come to the district, and increase the cultural and artistic offerings in it [38].

It is expected that mobility will be improved in the three districts based on an integral vision of the system. This includes the prevalence of the roles of pedestrians and cyclists, which are associated with an artist's profile and other profiles related to creative, innovative people and other participants of the Orange Economy. It also involves the low impact on the environment and the Sustainable Mobility Center that promotes quality public transport and coverage, as well as the change in the traditional paradigm of mobility.

In addition, with urban transformation plans, districts tend to create new public spaces and/or rehabilitate existing ones and pedestrianize streets and areas that used to be exclusively for vehicles, thus increasing the effective public space and, in turn, improving pedestrian conditions and including new mobility actors such as cyclists. According to Borèn et al. [39], there is a need for "new institutional arrangements between actors and institutions underpinning the local performance of creative economy" regarding urban mobility policies that interconnect people within the city. Public policies for urban mobility in Medellin started to build an intra-urban connectedness prioritizing sustainable mobility systems, such as public electric buses and metro systems.

The Districts of Ruta N and Perpetuo Socorro (planned or induced) are related to the renovation of depressed urban areas, while the District of El Poblado (spontaneous)

adapted the existing architecture without major spatial transformations. Therefore, in the three districts, one of the recent phenomena is evidenced by the rehabilitation of pre-existences and urban renewal processes at different scales. This allows for transforming the district without having to implement models and theories of tabula rasa, typical of modern urbanism plans of the 20th century. In addition, innovation districts usually house properties of historic value, which could become a component of a district's identity [40].

*4.4. Case Study Barranquilla: Description of the Genesis from the Dynamics That Preceded Its Spontaneous Formation and Subsequent Planning Processes*

According to the definition of creative districts, these refer to the conformation of intra-urban territorial nodes with a representative socio-cultural attribute, where organizations, sources of employment, and infrastructure dedicated to creative sector activities are gathered, and which drag a high concentration of artists [41]. Based on this definition, the attribute recognized for the city of Barranquilla (Special Industrial and Port District) is the cultural festival of Colombia, the El Carnival of Barranquilla, which UNESCO declared a World Heritage Site in 2003. This is due to its historical, cultural, and social importance as a channel for preserving its cultural and heritage expressions framed in two essential areas of the city.

For the Metropolitan Area of Barranquilla, two spontaneous polygons have been identified, which are driven by urban planning processes, leading to the strengthening and consolidation of their socio-demographic networks and economic dynamics, which ensure their functionality in an articulated manner with the pre-existing urban structure. They are Barrio Abajo (Carnival Creative Industry) and "Corredor Universitario" [University Corridor] (Gradual Convergence of Educational Facilities). Likewise, a similar future scenario can be predicted for the project District 4.0 or Center for Events and Gran Malecón del Río "Puerta de Oro" (recycled soils intended for new facilities for events, leisure, and tourism). These projects are the great city goal to concentrate different clusters based on the Orange Economy and economic activities supported by creativity and innovation, which are highly compatible with residential and commercial uses that will emerge with the inevitable real estate development, yet to be defined.

Finally, given the divergence of different ways to approach the concept of a creative district for the city of Barranquilla, the existence of a fourth Natural Cultural District is also identified. It is located in the heritage neighborhood of Barrio El Prado, which is its area of influence, and it is the sector of the creative cultural industries relevant to cultural tourism that emphasizes architecture, museums, and galleries immersed in the city's heritage neighborhoods. Moreover, several urban planning instruments have arisen after the declaration of the Assets of Cultural Interest (BIC, for its Spanish acronym). These instruments are framed in Resolutions of the national order and Decrees of the District order that seek to protect, recover, and revitalize the heritage and cultural zone under the process of abandonment and experiencing processes of underutilization and deterioration.

Among the documents comprising the review of public policy, seven support the research topic. They include the Development Plan Soy Barranquilla 2020–2023 (I am Barranquilla 2020–2023) and the Cultural and Vibrant City policy. The first one was approved on 22 May 2020 by the District Council of Barranquilla and contains a competitive and innovative city policy that aims to create a favorable environment for entrepreneurship and strengthening of SMEs (Technological Initiatives and Creative Industries). Additionally, it is supported by the programs Ecosystems for Productivity and Innovation and Barranquilla 2100. The second one, the Cultural and Vibrant City policy, establishes the promotion and strengthening of the city's cultural manifestations with the Cultural and Creative Industries Program [42].

This plan is framed by Decree 0447 of 29 December 2019, which delimits the Orange Development Area ADN Barrio Abajo, and Decree 0117 of 29 July 2005 [43,44], which adopts the Partial Plan for the protection, rehabilitation, recovery, and revitalization of the Historic Center of the District of Barranquilla. This first public policy frame-

work normalizes physical spatial relations and stimulates the cultural industry in the exploration area.

District 4.0, Center for Events and Gran Malecón del Río "Puerta de Oro": This is where facilities, public spaces, Gastronomy trade, and the Exhibition Center assigned for Barranquilla city converge, aiming to reconnect the city with the Rio Grande de la Magdalena. This required the implementation of various planning instruments such as partial plans, sectoral plans for water infrastructure, public space, mobility, and heritage conservation, among others. Then, in the medium term and dynamically, it was possible to develop a large-scale city project that, in addition to avoiding the expansionist tendency, is consistent with its industrial tradition [45].

Furthermore, all this complied with the emphatic provision of the current Land-use Planning (POT, for its Spanish acronym), Decree 0212 of 2014, to reorient the city towards a more compact, efficient, inclusive, and biodiverse model of occupation [46] through four functional units for its management. The first unit covers the Gastronomic Sector, which combines landscape treatments, recreational areas, and a square to increase the vitality of the area and the use of environmentally sensitive areas. Functional Unit 2 includes the Recreational Sector, which contributes to the green planning of the general public space system and has been increasing its urban tree planting effect, thus counteracting to some extent the effects of climate change, establishing new shadow areas for public space, and improving urban habitability conditions.

Orange Development Areas, ADN 1 and 2 in Barrio Abajo neighborhood: This is an area assigned for the Creative Industries of the Carnival where some important places are located, such as the Édgar Rentería Baseball Stadium, the Amira de la Rosa Theater, and the French Alliance; also, the Metropolitan Cathedral of Barranquilla and the Plaza de la Paz, the historic center of the city, the Parque Cultural del Caribe [the Caribbean Cultural Park], the Biblioteca Piloto del Caribe [the Pilot Library of the Carib-bean], and the Museum of the Caribbean.

On the other hand, among the major transformations planned for the area are the changes in densities. These have a high impact due to the changes in heights, with 8 stories for the heritage sector, 16 stories for consolidation treatment, and 40 stories for redevelopment and reactivation treatments. Thus, altering the current urban landscape of one-story buildings located throughout the neighborhood corresponds to approximately 84.09% of the total area [46]. This proposal will undoubtedly alter the urban morphology that has been able to adapt to the new activities historically recorded and that is recognized by its formal diversity of organic patterns and irregular shapes, which are the result of the operating urban conditions.

In addition to these economic and social changes, road infrastructure transformation projects involve the main structuring axes of motorized mobility, representing the perimeter mobility and access to the metropolitan area and its regional connection. In this way, the axes that represent the internal mobility system of the city, as well as the new infrastructures of the public mobility system, Transmetro, transform the urban landscape and provide large infrastructures. Among them are the boarding stations in their longitudinal connection to and from Via Puerto Colombia (Cartagena City) to Puente Pumajero (Rio Magdalena Port and Santa Marta City) in their strategic mobility with the Caribbean region.

Natural Cultural District El Prado and Alto Prado Neighborhoods: This is considered the area where Cultural Facilities converge in a Heritage Context. Therefore, it represents the Barranquilla aesthetics, as it concentrates on the hotel area and multiple technical and higher education service providers. Also, its heritage relevance has led to the emergence of a wide range of services, among which are tourism, finance, and trade. Moreover, its transformations occur simultaneously with changes in urban morphology, which is recognized by its second and third patterns (mixed and reticular), with interruptions by discordant urban elements such as large-scale trade, industry, and urban-scale educational activities.

The radial urban grid of the neighborhood facilitates the city's expansion to larger areas of the territory. And the construction of boulevards is a form of direct relationship with the environment. However, its main feature of flexibility is the change in the use of its mansions, which is activated by a new perspective of its local to urban scale, leaving architectural obsolescence at high risk. There might also be an increase in abandoned areas due to residual results from the re-configuration of plots and blocks that sacrifice patios and front gardens, thus degrading the topsoil and opening the possibilities for adjustments in the stratification of the urban layout.

The area incorporates two heritage zones declared by the national and local government, thus assigning the El Prado neighborhood and part of its area of influence as Cultural Interest Assets, covered by the Special Management and Protection Plan (PEMP) and the urban planning regulations contained in the Land-use Planning (POT). These regulations seek to preserve the heritage, architecture, and environment of this important neighborhood in Barranquilla, which is recognized as one of the first urban developments in Latin America. However, these neighborhoods face complexities such as changes in demands of diverse economic activities, displacement of their inhabitants to other parts of the city, and socio-urban degradation.

## 5. Discussion

After analyzing the case studies, it can be corroborated that the innovation districts in Bogota, Medellin, and Barranquilla partially comply with the characteristics and socio-spatial conditions stated in the literature reviewed. In terms of urban design, the case studies have shown that creative and innovation districts, to a greater or lesser extent, can be considered as physically compact and accessible regarding public transportation, with a complex and vibrant urban organization (formed by a mix of uses whether residential, commercial, for services, culture or entertainment); and physically hyperconnected with its immediate local surroundings. This is the result of the conditions of centrality that most of them have, firstly, because they are located in areas within the historic center of these cities and, secondly, because they correspond to emerging centralities, mainly due to the relocation of activities from the traditional center.

In order to limit negative externalities before and during the implementation of the districts in the different cities, it is essential to incorporate public leadership in their development, which favors inclusive and sustainable growth. This is particularly to limit gentrification and other negative urban phenomena resulting from strategies that generate capital gains and real estate rents.

In the case of Medellin and Bogota, strong public leadership is needed in the planning phase for an adequate transformation of the former industrial areas of the cities. Such leadership should allow coordination with the local government and actors in the area to form a quadruple helix (academia, public administration, private sector, and representative groups of citizens. Moreover, this helix should assume the construction of long-term objectives for the districts' urban, social, and economic sustainability, as well as define the general and functional urban structure, incorporating strategies for inclusion, social integration, and sustainability. In summary, all cases need urban renewal, which implies the modernization of the general urban systems like roads, public space, and parks by integrating the future and structuring mobility projects.

Something common in all three cases is that the regulations begin to glimpse aspects that need to be considered in the innovation districts. For example, urban structuring factors, such as land use and mobility, have begun to require transforming the environment in public spaces and the emergence of new facilities. In turn, these districts, which have new uses and are experiencing the contemporary dynamics of the creative city, are required to be linked to the mass transportation system.

In the three case studies, there is expected to be an urban and architectural renewal process, including the recycling of heritage structures or structures of architectural value that preserve part of the identity and memory of the place. In consequence, these processes

encourage the emergence of new facilities for the city and the district, which generally become highly important and symbolic urban landmarks. It is also expected that the ecological structure will be strengthened through quality urban interventions that consider the vegetation component in their designs in order to minimize heat island effects and pollution from air emissions.

Likewise, mobility is recognized as a fundamental part of these types of urban renewal processes. It also guarantees the connectivity and accessibility of these sectors with the rest of the city, giving them added value for the development of multiple activities and urban uses, and thus making them attractive for new talents for residence, commerce, and cultural activities, among others.

From this panorama regarding the growth and consolidation of the creative industries in the three cities, it is concluded that the cultural and patrimonial aggregate attributes an inherent revitalization to the urban dynamics that provide guarantees for the development of these activities. However, to counteract the effects of monofunctional areas of the orange industry, it is advisable to complement and reinforce the concept of the Districts around housing. It also includes the urban form of the neighborhoods, whose size favors the concentration of agglomeration economy for creative industries, thus guaranteeing accessibility and permeability at different scales. This, in turn, requires attention to the growth of green areas and the planning of mobility infrastructure to meet the needs of both pedestrians and public and private transportation.

As a whole, the identified polygons reveal some approaches that must be addressed for the continuity and success of the application of public policies both for the recognition and the allocation and implementation of current and future development plans, like the National Development Plan mentioned above, that consider the diversity of knowledge, the plurality of the ways of emerging and consolidating. These aspects are:

Spatial delimitation of Innovation Districts: it demonstrates the relevance of the construction of instruments and tools for planning and evaluating these special areas. These areas require analysis studies that incorporate both economic and social projections aimed at guaranteeing the recovery and/or redevelopment, thus reducing the impacts on the urban space transformation and benefiting local promoters who contribute to sustainability through the redefinition of trades and the promotion of entrepreneurship.

It should be considered that induced ADNs are allocated in unused spaces or depressed areas of the city that are projected as new focal points for economic, social, and cultural development despite the efforts to relocate the population and the high investment costs for their implementation. On the other hand, spontaneous ADNs create high densities of facilities and a high number of practices related to the area [3], causing pressures on the urban infrastructure of public services and mobility that must be anticipated in the intervention actions.

The emergence of indices for measuring innovation in the local and national context: they seek to identify the aspects of the environment and the enabling conditions that encourage innovation from five pillars: institutions, human capital and research, infrastructure, market sophistication, and business sophistication. They are gathered in the Input Sub-Index and the Output Sub-Index, which concentrates on the results of innovative activities or positive externalities resulting from innovation and which can be seen in the production of knowledge, technology, and creative production [22]. Both indexes are to be applied before and after by means of feasibility simulations and prospective scenarios supported by trend data and monitoring the factors during their implementation and operation.

Mobility and transportation within CD and ID: districts in big and medium-sized cities require new mobility systems, promoting shared transportation and sustainable public transportation. In this sense, "the lack of integration between transport services and transport modes" [47] becomes a problem for urban mobility and urban planning, especially in districts that concentrate on services and commodities like CD and ID. This evidences the need for integrated public transportation systems that can meet the demand and are based on sustainable criteria.

Based on the discussions, the authors recommend some possible directions for future research in Colombia regarding CD, ID, and Urban Sustainable Development. First, CD and ID strategies and policies must be reviewed and connected with the National Development Plan 2022–2026, considering the government´s commitment to the advantage of the built, participatory, and inclusive city to strengthen intra-urban links, innovation, and creative-based economies [30]. It is important to examine the role of CD and ID in the national drive for innovation, creativity, and sustainable development.

## 6. Conclusions

This study aimed to understand if innovation and creative districts are new scenarios for sustainable urban planning. The research resulted in the following insights. Firstly, creative and innovation districts are an opportunity to configure public space and green spaces as a fundamental part of sustainable urban planning. Through them, it is possible to increase permeable surfaces and implement nature-based solutions such as urban drainage while contributing to the ecological structure and strengthening its capacity for eco-systemic services; all this impacts the quality of life and well-being of the districts and the city's inhabitants.

Secondly, the changes in urban morphology adapt according to the different plans in each district and city. These changes often respond not only to new uses but also to new public spaces and ecological connectivity. Additionally, it is very important to recognize that there has been a process of gentrification in different magnitudes within all the districts reviewed. Therefore, it is necessary to incorporate mitigation and reduction plans of subsequent action for this negative effect since guaranteeing the permanence of original residents also guarantees the life and appropriation of public space at different times of the day.

In sum, CD and ID can be considered as new scenarios for sustainable urban development because they require new urban structures to guarantee their functioning and, for this, modifications are required in the cities, which should be guided by a sustainable planning and management vision that promotes interconnected green space, a multi-modal transportation system, and mixed-use development. Diverse public and private partnerships should be used to create sustainable and livable communities that protect historic, cultural, and environmental resources. In addition, policymakers, regulators, and developers should support sustainable site planning and construction techniques that reduce pollution and create a balance between built and natural systems.

**Author Contributions:** Conceptualization, A.E.B.-V., L.M.E. and C.V.-M.; methodology, A.E.B.-V., L.M.E. and C.V.-M.; validation, A.E.B.-V., L.M.E. and C.V.-M.; formal analysis, L.M.E. and C.V.-M.; investigation, A.E.B.-V., L.M.E. and C.V.-M.; resources, A.E.B.-V., L.M.E. and C.V.-M.; data curation, A.E.B.-V., L.M.E. and C.V.-M.; writing—original draft preparation, A.E.B.-V., L.M.E. and C.V.-M.; writing—review and editing, A.E.B.-V.; visualization, A.E.B.-V., L.M.E. and C.V.-M.; project administration, A.E.B.-V., L.M.E. and C.V.-M.; funding acquisition, A.E.B.-V., L.M.E. and C.V.-M. All authors have read and agreed to the published version of the manuscript.

**Funding:** This research was funded by Universidad Pontificia Bolivariana grant number 606C-08/20-28 And The APC was funded by Universidad Pontificia Bolivariana.

**Institutional Review Board Statement:** Not applicable.

**Informed Consent Statement:** Not applicable.

**Data Availability Statement:** Data was published by co-researchers Juan Eduardo Chica–Mejía, Luis Miguel Ríos Betancur and Jairo Eduardo Galvis Bonilla at ACE: architecture, city and environment. http://doi.org/10.5821/ace.17.51.11862.

**Acknowledgments:** The authors would like to dedicate this work to Walberto Badillo (RIP), who participated at the beginning of the research project and sadly passed away from COVID-19; his work and reflections are echoed in this paper.

**Conflicts of Interest:** The authors declare no conflicts of interest.

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
