# Peer review of "Are Innovation and Creative Districts New Scenarios for Sustainable Urban Planning? Bogota, Medellin, and Barranquilla as Case Studies"

_sustainability, doi:10.3390/su16073095_

Round 1

Reviewer 1 Report

Comments and Suggestions for Authors

The paper is very descriptive, does not fully address what is presented in the title

The paper would benefit if the relation in paragraph between sustainable urban planning effects on mobility were to be stated better, as it does not emerge clearly.

In premises on the choices in   planning are just stated. It needs to be reviewed, with an introduction at least to present the decision-making process in the adoption of sustainable urban planning.

A better view of the effects on urban marketing policies  would require the consideration of dynamic mobility management. A useful reference on methodology can be found in

 F. M. M. Cirianni, G. Leonardi Analysis of transport modes in the urban environment: an application for a sustainable mobility system doi: 10.2495/SC060611

Cirianni, F., Leonardi, G., Iannò, D. (2021). Operating and Integration of Services in Local Public Transport. https://doi.org/10.1007/978-3-030-48279-4_142

Comments on the Quality of English Language

There are minor grammatical flaws, which require minor editing of English language.

Author Response

Dear reviewer, 

Thank you very much for your comments and suggestions. We have restructured the paper taking into consideration all the reviewer's comments and suggestions. 

We read the papers you suggested on transport and the urban environment and included two paragraphs on our paper since we consider it fit perfectly in our discussion. 

In the Word document attached: in response to reviewers, you will find more information about the changes we did.

Best regards

Reviewer 2 Report

Comments and Suggestions for Authors

Dear Authors,

The article raises an important topic, and the case of Colombian cities is essential. Nevertheless, while reading the manuscript, I noticed a few things.

The Introduction was combined with the Research Review, which is not a good solution. The whole thing is relatively superficial. This chapter should be expanded - what discourses in this area are conducted in world literature? What is the authors' new achievement in this context, and why is this article important?

 The Methods chapter lacks many specifics. It is written too descriptively.

The figures in the Results section are illegible.

The discussion is superficial and does not refer to the findings of world literature.

The conclusions are too long. Conclusions should not cite other publications.

The list of references is not too long, considering the topic and research problem of the article.

I appreciate the Authors' contribution to the research, but the article needs refinement.

Author Response

Dear reviewer, 

In response to your comments, the authors restructured the paper and include the following.

  1. We separated the Literature Review from the Introduction to have a better structure for the paper and to expand the concepts reviewers suggested. The introduction was rewritten to include the authors' new achievements in this context and the importance of the article.
  2. Methods were reorganized to make it clear for readers and to show in a better way the research process.
  3. We change the image resolution for illegibility.
  4. The conclusions were rewritten and citations were not included.
  5. References were expanded with suggestions made by some of the reviewers and other publications that contributed to the discussion.

In the attached Word document you can find more information about all the changes we made considering all reviewers' comments,

Best regards

Reviewer 3 Report

Comments and Suggestions for Authors

It is a paper that could not need improvements to be published in my opinion. The structure is simple and well organized, IMRAD well applied and ideas developed cleanly, very readable, and well balanced in all its parts. The bibliography is good, UP-To-Date, and well-selected.

BUT

The only thing I see is that I don't see any link to Sustainability in the paper, or, at least it is hidden and must be valorized. I guess there is much room for improvement on this side, maybe talking about Green Sustainability, for sure, and more directly about Social and Economic Sustainability from the beginning, as a core part of the paper.

This last comment can be the only reason to me for not publishing directly the paper but looking into the Policy of the editor for this journal.

The paper is very good, it is just need this point to be added.

Author Response

Dear reviewer, 

Thank you for your comments and suggestions, we have made more evident all the concepts of Green Sustainability and Sustainable Urban Development as you suggested. The paper changed a lot due to other reviewers' suggestions, but we maintained the discursive line, adjusting concepts and critical analyses that were requested.
We thank you for your review, in the attached file you will see the response to all comments in general. 

Best regards

Reviewer 4 Report

Comments and Suggestions for Authors

The scope of this research is interesting and relevant to urban planning. The paper is well written. Just a few minor remarks:

The proofreading of the manuscript is required by the authors themselves to eliminate redundancy.

Line 81 - please consider revising, meaning unclear

Please consider revising the manuscript to organise the comparison of the three case studies in a more legible manner. Maybe a table could be an option, or similar subsections in each case study.

Lines 532-541 - please consider revising to avoid repetition

The conclusions should be concise. Please consider transfering some of the paragraphs to the section with results or discussion

Author Response

Dear reviewer, 

Thank you for your comments. We reviewed the paper to eliminate redundancy and grammatical errors. 

The conclusion section was reorganized. You will find all the changes the authors made in the Word document. 

Best regards

Reviewer 5 Report

Comments and Suggestions for Authors

This manuscript (sustainability-2887585) aims to analyze three case studies of the cities of Barranquilla, Bogota, and Medellin in Colombia, seeking to characterize the creative and innovation districts. The methodology used is too much qualitative, resorting to the characterization of the polygons and the analysis of public policies. After reviewing the entire manuscript, I have several comments and suggestions as listed in the following:

- . The contents in the Abstract are too simple that potential readers can hardly grasp the important points of your research.

- . The full and clear explanation and definition of the term "orange economy" should be provided in the very first beginning of the Introduction Section.

- . The manuscript does not clearly propose a specific research question or hypothesis, which makes it difficult for readers to understand the main purpose and research focus of the manuscript. A good research paper should have a clear research question or hypothesis, which helps guide the reader to understand the main content and objectives of the paper.

- . The literature review section of the manuscript is relatively weak and does not fully cover important research in related fields, nor does it conduct an in-depth critical analysis of previous research. This results in insufficient background information in the paper, which affects readers' understanding of the whole paper.

- . The manuscript lacks clarity when describing the research methods, and does not explain in detail how the data were collected and analyzed, and why this method was chosen. This makes it difficult for readers to understand the research process of the paper and also affects their trust in the research results.

- . The study has involved a smaller sample size, which may have led to underrepresentation of the findings. To increase the reliability of the study, consideration should be given to expanding the sample size.

- . This study did not analyze the factors influencing the development of creative and innovative districts in depth enough and ignored other important influencing factors. This may result in research results that cannot accurately reveal the internal mechanisms and laws of the development of creative and innovative regions.

- . Generally speaking, its content and methods lack some discussion of the innovation of this manuscript.

- . The discussion part of the manuscript is too general and does not specifically point out the main findings and contributions of the research, nor does it provide a clear outlook on future research directions. A good conclusion section should summarize the main ideas and findings of the paper and provide specific recommendations and directions for future research.

- . Finally, I also suggest that some quantitative methods are necessary for the purpose of this research.

Comments on the Quality of English Language

Minor editing of English language required

Author Response

Dear reviewer, 

Thank you very much for your comments and suggestions, and your dedicated reading of our work. In the Word document, you will find all the changes we made to the paper to respond to your suggestions and the other reviewers. 

Best regards

Round 2

Reviewer 1 Report

Comments and Suggestions for Authors

If the paper has seen some improvements in the descriprion, I still have difficolutly in relating an answer to the main question of the title. I strongly advise to revise to better present the requested conclusions, which the authors would wish to present

Author Response

Dear reviewer, 

Please see the attachment,

Regards

Reviewer 2 Report

Comments and Suggestions for Authors

Dear Authors,

I have two minor comments that improve the manuscript:

1. Chapter 6. Future research agenda should be integral to the Discussion (Chapter 5). They should not be presented separately.

2. The inscriptions in Figure 1 are practically impossible to read.

Author Response

Dear reviewer, 

Regards

Reviewer 3 Report

Comments and Suggestions for Authors

The paper improved in many parts, at least the world sustainability has been added in many of these also. The quality of the scientific deepness is quite average, the methodology part is not well developed and not fully linked with other parts, literature review must be enriched with papers up-to-date and with more standing and significance.

Author Response

Dear reviewer,

Please see the attachtment 

Regards

Reviewer 5 Report

Comments and Suggestions for Authors

I have no further comments.

Author Response

(The authors gave the same response as above.)

Round 3

Reviewer 1 Report

Comments and Suggestions for Authors

The revised paper is improved, giving a clear exposition of the presented research. The case studies are well presented. I would advice for further developments to better define the relation between mobility and urban planning. 

Reviewer 3 Report

Comments and Suggestions for Authors

All the comments in the previous review have been sufficiently implemented.